# Feedback control of heart rate during treadmill exercise based on a two-phase response model

**Hanjie Wang** \*, **Kenneth J. Hunt**

rehaLab - the Laboratory for Rehabilitation Engineering, Division of Mechatronics and Systems Engineering, Department of Engineering and Information Technology, Institute for Human Centered Engineering HuCE, Bern University of Applied Sciences, Biel, Switzerland

\* hanjie.wang@bfh.ch

## Abstract

This work investigated automatic control of heart rate during treadmill exercise. The aim was to theoretically derive a generic feedback design strategy that achieves a constant input sensitivity function for linear, time-invariant plant models, and to empirically test whether a compensator $C_2$ based on a second-order model is more dynamic and has better tracking accuracy than a compensator $C_1$ based on a first-order model. Twenty-three healthy participants were tested using first and second order compensators, $C_1$ and $C_2$, respectively, during 35-minute bouts of constant heart rate treadmill running. It was found that compensator $C_2$ was significantly more accurate, i.e. it had 7% lower mean root-mean-square tracking error (1.98 vs. 2.13 beats per minute, $p = 0.026$), and significantly more dynamic, i.e. it had 17% higher mean average control signal power ($23.4 \times 10^{-4}$ m$^2$/s$^2$ vs. $20.0 \times 10^{-4}$ m$^2$/s$^2$, $p = 0.011$), than $C_1$. This improvement likely stems from the substantially and significantly better fidelity of second-order models, compared to first order models, in line with classical descriptions of the different phases of the cardiac response to exercise. These outcomes, achieved using a treadmill, are consistent with previous observations for the cycle ergometer exercise modality. In summary, whenever heart rate tracking accuracy is of primary importance and a more dynamic control signal is acceptable, the use of a compensator based on a second-order nominal model is recommended.

## Introduction

During exercise training, it can be beneficial to vary exercise intensity between two or more levels for varying durations: this so-called "interval training" has advantages compared to continuous training [1–3]. Because heart rate (HR) is a variable that is commonly used to set exercise intensity [4], this has inspired the development of accurate and robust HR control systems. In the present work, we employed a treadmill (TM) as the platform for HR control. Compared to other exercise modalities, e.g. cycle ergometry (CE), treadmill exercise has a relatively high energy expenditure at a given HR [5], especially at low to moderate intensities, cf. [6].

**Data Availability Statement:** All measurement/data files are available from the OLOS database. DOI:10.34914/olos:szica6xuenc23pe4ft6smihktq).

**Funding:** This work was supported by the Swiss National Science Foundation (Principal Investigator

KH, Grant Ref. 320030-185351). snf.ch The
funders had no role in study design, data collection
and analysis, decision to publish, or preparation of
the manuscript.

**Competing interests:** The authors have declared
that no competing interests exist.

Recent studies of feedback control of heart rate are mostly model-based and can be divided into two classes: several investigations [7–10] employed the nonlinear model proposed by Cheng *et al.* [11], while others did the feedback design using linear time-invariant models [12, 13]. Because the stability of nonlinear controllers is conditional [8, 10], the corresponding HR control studies mainly focused on theoretical stability and robustness analysis, while empirical verification used either simulation [7, 8] or experiments with very small numbers of participants [9, 10].

In contrast, recent studies with large and statistically meaningful test group sizes showed that a first-order, linear, time-invariant (LTI) model can be used to design LTI compensators that achieve impressive HR control performance: a HR control study on healthy participants ($n = 30$) achieved a mean root-mean-square heart rate tracking error (RMSE) of 2.96 beats per minute (bpm) [14]; another study on healthy participants ($n = 25$) that compared linear HR control performance on TM and CE modalities achieved a mean RMSE of 2.85 bpm on the treadmill and 3.10 bpm on the cycle ergometer [15]. Moreover, a systematic comparative study with sample size $n = 16$ showed that HR control performance with linear and nonlinear controllers was not significantly different; in fact, the nonlinear controller had worse performance at low speeds [16].

With a view to further improving the performance of HR control based on linear models, a second-order model structure was investigated in our previous treadmill-based model identification study [17]. This structure was motivated by physiological knowledge regarding human HR response to exercise that identifies three phases: Phase I is a small but immediate response, Phase II is a large and slower component that comprises the main part of the overall response, and Phase III is an ultra-slow drift that can occur above the anaerobic threshold [18]. Since, in feedback control, very slow disturbances can be effectively eliminated by integral action, Phases I and II are significant for feedback design: this motivated our investigation of two-phase, i.e. second order, models. The comparative analysis with $n = 22$ showed that the second-order model structure achieved significantly better fit due to the explicit delineation of the fast and slow dynamic components of HR response. Furthermore, the improvement is not limited to treadmills: a similar model identification study using a cycle ergometer ($n = 26$) gave consistent results [19].

The improved fidelity of second-order models, due to the inclusion of fast and slow dynamics, leads to the hypothesis that compensators based on these models could be more dynamic and thus might deliver more accurate heart rate tracking. The concept of a controller being more or less dynamic is quantified in this work using the average power of changes in the control signal (the treadmill speed command; see Eq (32) in the sequel). This hypothesis was experimentally tested in a pilot study with 10 participants [20]. It was found that compensators derived using second-order models, denoted $C_2$, were significantly more dynamic than compensators based on first-order models, $C_1$. However, there were no significant differences in heart rate tracking accuracy between $C_1$ and $C_2$. This outcome was likely due to important differences between the sensitivity functions for the two compensators, differences in the reference prefilter design, and the sample size being too small to allow detection of differences, even when they exist. These limitations were addressed in a subsequent study using a cycle ergometer [19]: with $n = 26$ participants, controller type $C_2$ was found to be significantly more dynamic and more accurate than type $C_1$.

This analysis motivates further study of control performance of $C_2$ vs. $C_1$ using the treadmill modality. Here, we rectify the limitations of the treadmill pilot study [20], in the following way:

1. the feedback design was modified to make the input-sensitivity function constant over all frequencies, which in consequence made the other sensitivity functions flatter (less

peaking) and the nominal functions for $C_1$ and $C_2$ more similar; this required a new theoretical derivation, presented here, for the second-order case of a feedback compensator $C_2$ that gives a constant input sensitivity function;

2. the confounding effect of the reference prefilter was eliminated by considering only a constant HR reference signal;

3. statistical power was improved by increasing the sample size to be similar to previous studies that were well powered, viz. we used $n = 23$.

The aim of this work was twofold: (i) theoretical contribution—to develop a novel feedback design strategy that achieves a constant input sensitivity function for LTI plant models in general, and for the second-order case in particular; (ii) experimental contribution—to empirically test whether a compensator based on a second-order model is more dynamic and has better HR tracking accuracy than a compensator based on a first-order model.

## Materials and methods

### Controller design

The control structure used for this study has a standard form consisting of two parts (Fig 1): a feedback compensator $C(s)$ that adjusts the control signal $u$, i.e. the commanded speed of the treadmill, in dependence upon the error $e$ between filtered HR reference $r'$ and the HR measurement $z$; and a reference prefilter $C_{\mathrm{pf}}$ to manipulate the overall tracking response of the closed-loop system, i.e. the response from reference heart rate $r = \mathrm{HR}^*$ to actual heart rate $y = \mathrm{HR}$. A disturbance term $d$ models heart rate variability (HRV) and other sources of uncertainty, and $n$ represents measurement noise.

**Nominal plant model.** As detailed in our previous model identification study [17], the general form for the nominal plant $P_o$ is taken to be the strictly proper transfer function

$$P_o(s) = \frac{B(s)}{A(s)} : \ u \mapsto y \tag{1}$$

where $A$ and $B$ are polynomials with $A$ monic and their degrees satisfy $n_b < n_a$. The algebraic strictly-proper condition $n_b < n_a$ corresponds in the frequency domain to a transfer function that has low pass characteristics, viz. $\lim_{\omega \to \infty} |P_o(j\omega)| = 0$.

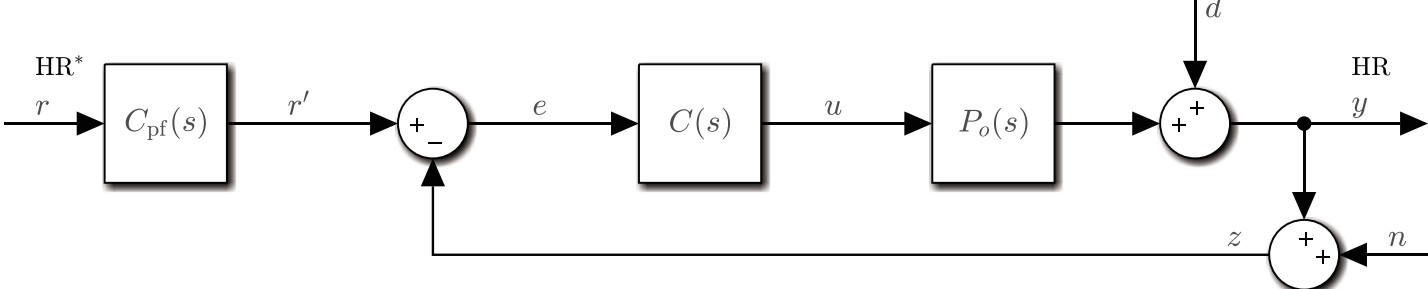

**Fig 1. Control structure of this study.** Nominal plant model $P_o(s)$, feedback compensator $C(s)$ and reference prefilter $C_{\mathrm{pf}}(s)$. The controlled variable $y$ is the heart rate (HR) and $r$ is the reference/target heart rate (HR\*). The control signal $u$ is the commanded speed of the treadmill. The terms $d$ and $n$ respectively represent disturbances and measurement noise.

In the sequel we solve the general compensator problem for the generic plant Eq (1) and then specialise the solution to two instances of the plant: a first-order form

$$P_1(s) = \frac{k_1}{\tau_1 s + 1} = \frac{\dfrac{k_1}{\tau_1}}{s + \dfrac{1}{\tau_1}},$$
(2)

where $B(s) = k_1/\tau_1$ and $A(s) = s + 1/\tau_1$, and a second-order form

$$P_2(s) = \frac{k_2}{(\tau_{21} s + 1)(\tau_{22} s + 1)} = \frac{\dfrac{k_2}{\tau_{21}\tau_{22}}}{\left(s + \dfrac{1}{\tau_{21}}\right)\left(s + \dfrac{1}{\tau_{22}}\right)}$$
(3)

where $B(s) = k_2/(\tau_{21}\tau_{22})$ and $A(s) = (s + 1/\tau_{21})(s + 1/\tau_{22})$. $k_1$ and $k_2$ are steady-state gains and $\tau_1$, $\tau_{21}$ and $\tau_{22}$ are time constants. The first-order model Eq (2) effectively combines Phase I and Phase II dynamics into a single time constant $\tau_1$, while the second-order model Eq (3) has separate time constants $\tau_{21}$ and $\tau_{22}$ for each phase of the response.

**Compensator and sensitivity functions.** The compensator $C$ is a rational function that is constrained to include integral action,

$$C(s) = \frac{G(s)}{H(s)} = \frac{G(s)}{sH'(s)} : \quad e \mapsto u,$$
(4)

where the $G$, $H$ and $H'$ have degrees $n_g$, $n_h$ and $n_{h'}$, respectively, and $H$ is taken to be monic. The integral term $1/s$ is introduced to compensate the very slow Phase III dynamic of HR response.

In the following, we consider the transfer function $C$ to be merely proper (rather than strictly proper) with $n_g = n_h$. This is to prevent the input sensitivity function $U_o$ (Eq (7), below) from rolling off to zero with increasing frequency, in line with our design goal to maintain a constant $|U_o(j\omega)|$ for all frequencies: since $U_o = C/(1 + CP_o)$, and since $P_o$ is by definition strictly proper (whence $\lim_{\omega \to \infty} |P_o(j\omega)| = 0$), $U_o \approx C$ at high frequency. With a proper $C$, $U_o$ must also be proper and $|U_o(j\omega)|$ therefore remains finite.

For the feedback system of Fig 1, the principal sensitivity functions—sensitivity function $S_o$, complementary sensitivity function $T_o$ and input-sensitivity function $U_o$—are defined classically [21], as

$$S_o(s) = \frac{1}{1 + C(s)P_o(s)} : \quad d \mapsto y,$$
(5)

$$T_o(s) = \frac{C(s)P_o(s)}{1 + C(s)P_o(s)} : \quad r', n \mapsto y,$$
(6)

$$U_o(s) = \frac{C(s)}{1 + C(s)P_o(s)} : \quad d, r', n \mapsto u.$$
(7)

Using the rational forms for the plant $P_o$, Eq (1), and compensator $C$, Eq (4), the input-sensitivity function can be written

$$U_o(s) = \frac{AG}{AH + BG} = \frac{AG}{AsH' + BG} = \frac{AG}{\Phi} \tag{8}$$

where $\Phi = AH + BG$ is the closed-loop characteristic polynomial.

**Compensator synthesis for constant input sensitivity: General solution.** Since the feedback design goal is to achieve an input-sensitivity magnitude that is constant for all frequencies, it follows that the closed-loop poles (the roots of $\Phi$) must be completely cancelled by the numerator polynomial $AG$ in Eq (8). This can be achieved by firstly cancelling the plant poles by including $A$ in the compensator numerator, i.e. by setting $G = AG'$, giving

$$U_o(s) = A^2 G'/(A(H + BG')) = AG'/(H + BG'). \tag{9}$$

Second, the remaining closed-loop poles (the roots of $H + BG'$) are also placed at the open-loop poles by setting $H + BG' = A$ to give $U_o = G'$. Considering now that we have set $\Phi = A^2$ with degree $n_\varphi = 2n_a$, and that the numerator $AG = A^2 G'$ of Eq (8) must have this same degree, it follows that $2n_a + n'_g = 2n_a$, thus $n'_g = 0$ and $G'$ is therefore a constant, denoted $g'_0$. Consequently,

$$H + g'_0 B = A \Leftrightarrow sH' + g'_0 B = A \tag{10}$$

and

$$U_o = g'_0 \tag{11}$$

which, as desired, is a constant value. The compensator parameters are obtained by solution of Eq (10) for $g'_0$ and $H'$; since the compensator is proper ($n_g = n_h$), and since $G = g'_0 A$, $n_g = n_h = n_a$ and

$$n'_h = n_h - 1 = n_a - 1. \tag{12}$$

The controller solution derived above is valid for the generic nominal plant model $P_o = B/A$, Eq (1), without any restriction on the plant order. In summary, with the constraint $G = g'_0 A$, the compensator Eq (4) that achieves a constant $U_o = g'_0$, Eq (11), is given by

$$C(s) = \frac{G(s)}{H(s)} = \frac{g'_0 A}{sH'(s)} \tag{13}$$

where constant $g'_0$ and polynomial $H'$ are the unique solution of the polynomial Eq (10).

The cancellation strategy also leads to simplifications in the expressions for $S_o$ and $T_o$. From Eqs (5) and (6), and using $AH + BG = A^2$ and $G = g'_0 A$,

$$S_o(s) = \frac{1}{1 + C(s)P_o(s)} = \frac{AH}{AH + BG} = \frac{H}{A}, \tag{14}$$

$$T_o(s) = \frac{C(s)P_o(s)}{1 + C(s)P_o(s)} = \frac{BG}{AH + BG} = \frac{g'_0 B}{A}. \tag{15}$$

**Compensator synthesis for constant input sensitivity: First and second order solutions.** In the following, the general solution is specialised to first and second-order plants described by Eqs (2) and (3), respectively.

**First-order solution.** In the first-order case with $n_a = 1$, Eq (12) gives $n'_h = 0$ which in turn leads to the trivial solution $H' = 1$ and $H = s$ ($H$, and therefore also $H'$, were assumed at the outset to be monic). Eq (10) then reads

$$s + g'_0 \cdot \frac{k_1}{\tau_1} = s + \frac{1}{\tau_1} \tag{16}$$

which, by equating coefficients, has the solution

$$g'_0 = \frac{1}{k_1}. \tag{17}$$

For first-order plants, the compensator is therefore (see Eq (13))

$$C_1(s) = \frac{g'_0 A}{sH'(s)} = \frac{\frac{1}{k_1}\left(s + \frac{1}{\tau_1}\right)}{s} \tag{18}$$

and the corresponding sensitivity functions are, using Eqs (11), (14) and (15),

$$U_1 \quad = \quad g'_0 = \frac{1}{k_1}, \tag{19}$$

$$S_1(s) \quad = \quad \frac{H}{A} = \frac{s}{s + \frac{1}{\tau_1}}, \tag{20}$$

$$T_1(s) \quad = \quad \frac{g'_0 B}{A} = \frac{\frac{1}{\tau_1}}{s + \frac{1}{\tau_1}}. \tag{21}$$

**Second-order solution.** In the second-order case with $n_a = 2$, Eq (12) gives $n'_h = 1$, thus $H' = s + h'_0$ and $H = s(s + h'_0)$. A solution is then sought for Eq (10),

$$s(s + h'_0) + g'_0 \cdot \frac{k_2}{\tau_{21}\tau_{22}} = \left(s + \frac{1}{\tau_{21}}\right)\left(s + \frac{1}{\tau_{22}}\right), \tag{22}$$

which is obtained by straightforward algebraic manipulation as

$$g'_0 = \frac{1}{k_2}, \; h'_0 = \frac{1}{\tau_{21}} + \frac{1}{\tau_{22}}. \tag{23}$$

For second-order plants, the compensator is therefore (see Eq (13))

$$C_2(s) = \frac{g'_0 A}{sH'(s)} = \frac{\frac{1}{k_2}\left(s^2 + \left(\frac{1}{\tau_{21}} + \frac{1}{\tau_{22}}\right)s + \frac{1}{\tau_{21}\tau_{22}}\right)}{s\left(s + \frac{1}{\tau_{21}} + \frac{1}{\tau_{22}}\right)}. \tag{24}$$

From Eqs (11), (14) and (15), the corresponding sensitivity functions are

$$U_2 \quad = \quad g_0' = \frac{1}{k_2}, \tag{25}$$

$$S_2(s) \quad = \quad \frac{H}{A} = \frac{s\left(s + \dfrac{1}{\tau_{21}} + \dfrac{1}{\tau_{22}}\right)}{\left(s + \dfrac{1}{\tau_{21}}\right)\left(s + \dfrac{1}{\tau_{22}}\right)}, \tag{26}$$

$$T_2(s) \quad = \quad \frac{g_0' B}{A} = \frac{\dfrac{1}{\tau_{21}\tau_{22}}}{\left(s + \dfrac{1}{\tau_{21}}\right)\left(s + \dfrac{1}{\tau_{22}}\right)}. \tag{27}$$

The Eqs (18) and (24) for $C_1$ and $C_2$ show that the compensators comprise very simple expressions that depend only on the nominal plant parameters: no additional tuning parameters are otherwise involved.

**Compensator calculation.** For calculation of the compensator coefficients, nominal first and second-order plant parameters obtained from our previous identification study [17], were used: these models are averages obtained from 22 individual first and second order models from 11 participants (for each participant there were two first-order models and two second-order models), with values $k_1$ = 28.57 bpm/(m/s), $\tau_1$ = 70.56 s and $k_2$ = 24.70 bpm/(m/s), $\tau_{21}$ = 18.60 s, $\tau_{22}$ = 37.95 s. Substituting in Eqs (18) and (24), the compensators are

$$C_1(s) = \frac{0.0350s + 0.000496}{s} \tag{28}$$

and

$$C_2(s) = \frac{0.0405s^2 + 0.00324s + 0.0000574}{s(s + 0.0801)}. \tag{29}$$

The magnitude plots of the resulting closed-loop sensitivity functions for the pairs $P_1$, $C_1$ and $P_2$, $C_2$ are illustrated in Fig 2. It can be seen that the input sensitivity magnitudes are constant, according to Eqs (19) and (25), and that the complementary sensitivity function for the second-order case rolls off twice as fast as for the first-order case (40 dB/decade vs. 20 dB/decade, Eq (21) vs. Eq (27)).

Finally, the reference prefilter $C_{pf}$ was designed to make the overall closed-loop transfer function ($r \mapsto y$, see Fig 1) equal to a second-order transfer function denoted $T_{cl} = C_{pf} \cdot T_o$, where $T_o$ is the nominal complementary sensitivity function in each case. $T_o$ had critical damping and 150 s rise time (detailed in [14]). The transfer function of $C_{pf}$ was then

$$C_{pf}(s) = T_o^{-1}(s) \cdot T_{cl}(s) : \quad r \mapsto r'. \tag{30}$$

## Experimental design

Twenty-three healthy participants were recruited for this study with ages between 23 years and 57 years, mean body mass 71.4 kg and mean height 1.77 m. All participants were regular exercisers (at least 3 times a week, 30 minutes each time), non smokers and free from cardiovascular disease or musculoskeletal complaints. The study was approved by the Ethics Committee of

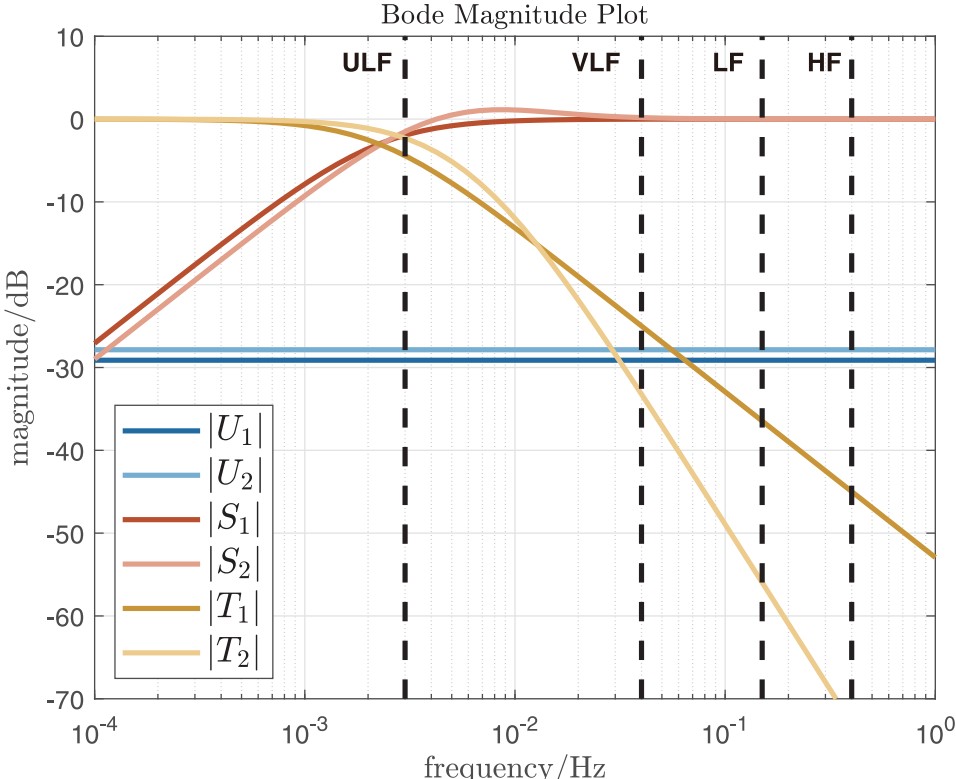

**Fig 2. Closed-loop frequency responses.** Input-sensitivity functions $U_{1,2}$, sensitivity functions $S_{1,2}$ and complementary sensitivity functions $T_{1,2}$ where the 1 and 2 subscripts denote the first and second-order plant/compensator combinations $P_1$, $C_1$ and $P_2$, $C_2$, respectively. The four vertical dashed lines at $0.00\bar{3}$ Hz, 0.04 Hz, 0.15 Hz and 0.4 Hz delineate the four frequency bands classically used for heart rate variability analysis [22]: ultra-low frequency (ULF), very-low frequency (VLF), low frequency (LF) and high frequency (HF).

the Swiss Canton of Bern (Ref. 2019-02184) and participants provided written, informed consent prior to participation.

Each participant completed two feedback control tests on the treadmill, one with $C_1$ and one with $C_2$. The order of presentation of each compensator ($C_1$ first or $C_2$ first) was changed in sequence for each participant to counterbalance possible order-of-presentation effects. Each test was conducted on a separate day and consisted of a 10-minute warm up, 10-minutes rest and a 35-minute formal measurement (Fig 3). During the warm up and formal measurement phases, the treadmill speed was controlled by the HR compensator ($C_1$ or $C_2$) to regulate the HR of the participant at a constant, mid-level target value, denoted $HR_m$, that was set to an intensity level at the border between moderate and vigorous using the prediction equation $HR_m = 0.765 \times (220 - \text{age})$ [23]. The 30-minute time interval from 295 s to 2095 s was used for outcome evaluation ("evaluation period" in Fig 3), except for participants P04 and P09 where the period 295 s to 1795 s was used (for both of these participants, there were measurement artefacts in the final five minutes of the tests).

## Equipment

We used a PC-controlled treadmill (model pulsar, h/p/cosmos Sports & Medical GmbH, Germany; Fig 4). The heart rate compensators were implemented in Simulink Desktop Real-Time

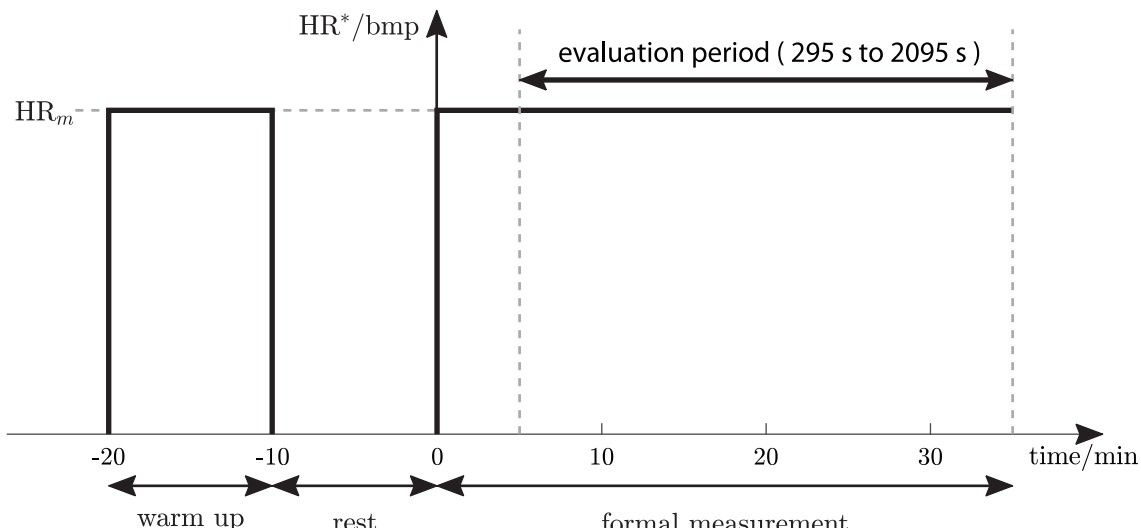

**Fig 3. Test protocol.** Target heart rate (HR*) for each feedback control test. $HR_m$ is an individually chosen mid-level heart rate.

(The MathWorks, Inc., USA) and speed commands were sent to the treadmill over a serial communication protocol. Heart rate was measured by a chest belt sensor (H10, Polar Electro Oy, Finland) and sent to Simulink through Bluetooth. Heart rate was sampled at 1 Hz while the compensators ran at a rate of 0.2 Hz (sample interval 5 s), hence the heart rate was down-sampled by averaging every five consecutive values.

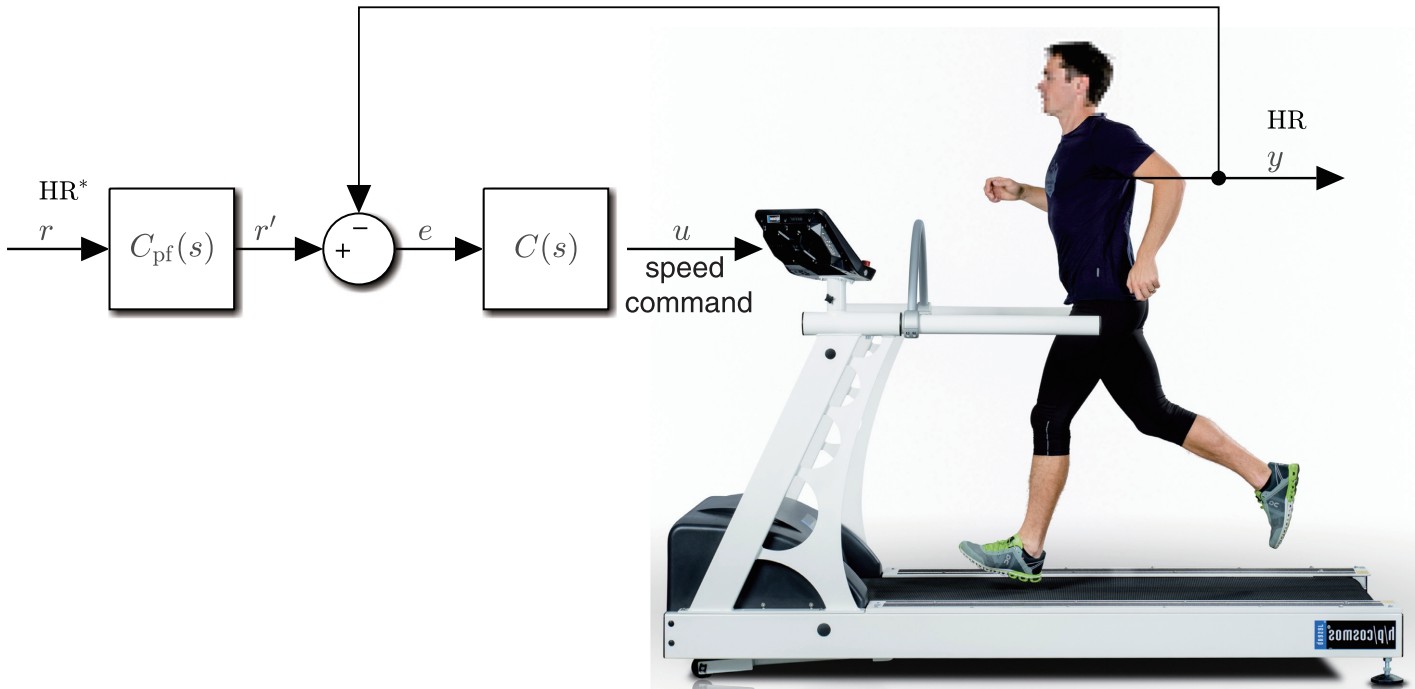

**Fig 4. Test equipment.** The computer-controlled treadmill used in this study—h/p/cosmos pulsar—embedded in the heart rate control feedback loop (cf. control structure in Fig 1). Adapted from hpcosmos.com.

## Outcome measures and statistical analysis

The accuracy of heart rate tracking was quantified using the root-mean-square error (RMSE) between measured HR and the nominal, simulated HR response. The intensity of the control signal, i.e. the treadmill speed command $u$, was evaluated by the average power of changes in this variable ($P_{\nabla u}$):

$$\text{RMSE} = \sqrt{\frac{1}{N}\sum_{i=1}^{N}(\text{HR}_{\text{sim}}(i) - \text{HR}(i))^2},$$ (31)

$$P_{\nabla u} = \frac{1}{N-1}\sum_{i=2}^{N}(u(i) - u(i-1))^2.$$ (32)

$\text{HR}_{\text{sim}}$ represents the simulated closed-loop HR response, i.e. $\text{HR}_{\text{sim}} = T_{\text{cl}} \cdot \text{HR}^*$, but since the target HR was constant during these experiments, $\text{HR}_{\text{sim}} = \text{HR}_m$ during the evaluation period: that is to say, the reference prefilter only played a role during the initial transient prior to the evaluation period and had no effect on the outcome calculation.

$i$ indexes the discrete sample instants and $N$ is the total number of samples during the evaluation period. As the evaluation period was from 295 s to 2095 s (Fig 3) and the sample rate of the compensator was 0.2 Hz, $N = 361$ (except for P04 and P09, where $N = 301$ as the evaluation period was from 295 s to 1795 s).

The hypothesis of this study was that, compared with $C_1$, $C_2$ would have better HR tracking accuracy, signified through a lower RMSE, and a more dynamic control signal, manifest by a higher $P_{\nabla u}$; thus, for both outcomes, one-sided hypothesis testing was done. Prior to hypothesis testing, a Kolmogorov-Smirnov test with Lilliefors correction was used to assess normality of sample differences. As all the differences were found not to significantly deviate from normality, testing was performed using paired t-tests. The significance level was set as $\alpha = 0.05$. Statistical calculations were implemented in the Matlab Statistics and Machine Learning Toolbox (The Mathworks, Inc., USA).

## Ethics statement

This research was performed in accordance with the Declaration of Helsinki. The study was reviewed and approved by the Ethics Committee of the Swiss Canton of Bern (Ref. 2019-02184). All participants provided written, informed consent to participate.

## Results

From the 23 pairs of data sets, measurements of 3 participants were excluded from the analysis for the following reasons: P08 reported feeling unwell during the second test; for P17, the treadmill speed was sometimes too low for the participant to be able to keep running; for P19, the HR response was affected by measurement artefacts. In the end, 40 valid data sets were obtained from 20 participants and the outcomes analysed.

For illustration, the measurements with the best (lowest), middle (closest to mean) and worst (highest) RMSE for the two compensators $C_1$ and $C_2$ are provided as representative data (Fig 5), while the overall results are summarised in Table 1 and Fig 6.

Overall, compensator $C_2$ was found to be significantly more accurate and significantly more dynamic than $C_1$, i.e. for $C_2$ mean RMSE was lower and mean $P_{\nabla u}$ was higher than for $C_1$: RMSE was 1.98 bpm ± 0.49 bpm vs. 2.13 bpm ± 0.35 bpm (mean ± standard deviation), $C_2$ vs. $C_1$, $p = 0.026$ (Table 1, Fig 6A); $P_{\nabla u}$ was $23.37 \times 10^{-4}$ m$^2$/s$^2$ ± $13.09 \times 10^{-4}$ m$^2$/s$^2$ vs.

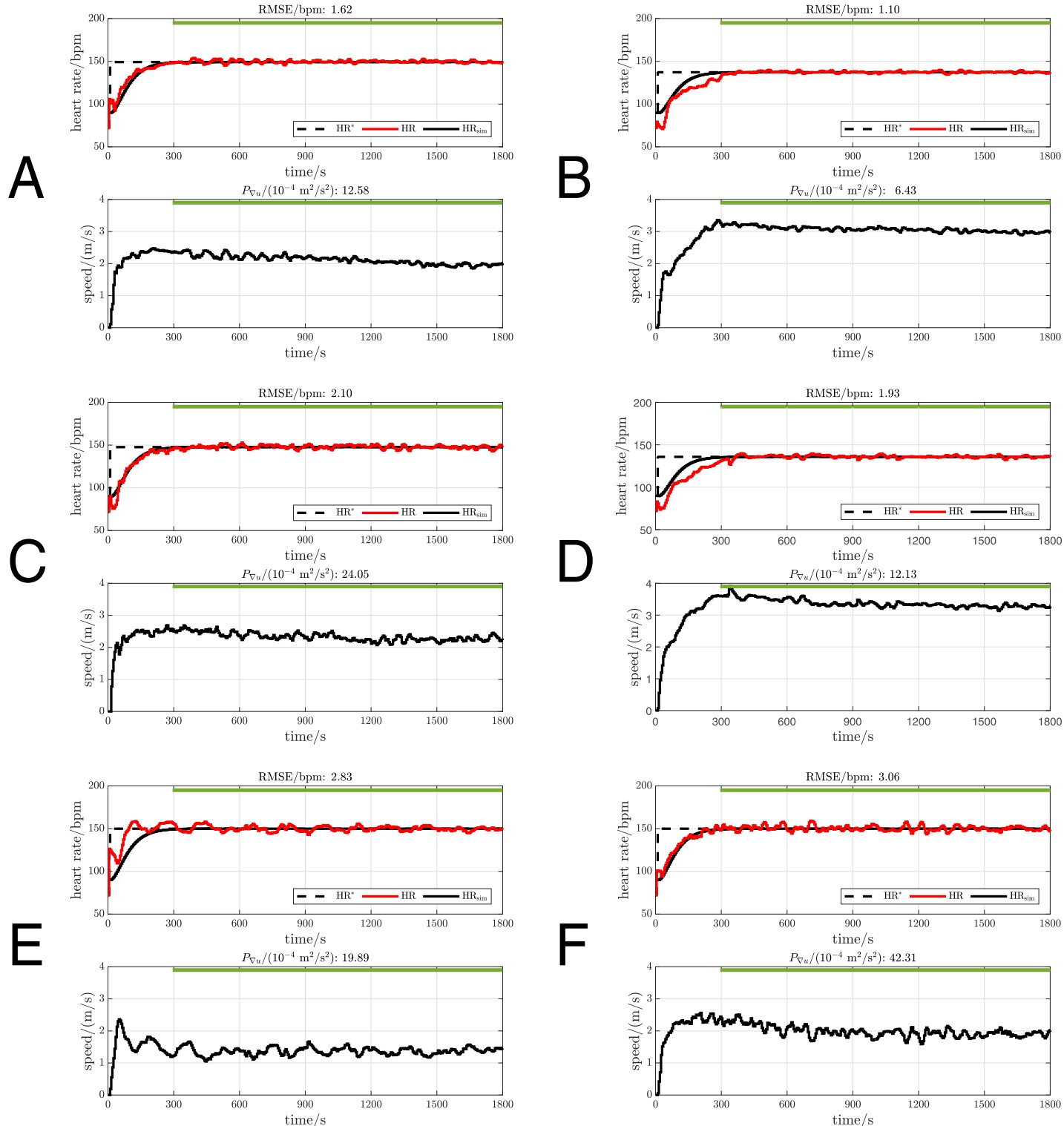

**Fig 5. Sample measurements with $C_1$ and $C_2$ that had the lowest, middle and highest RMSE.** The upper plot of each figure shows the reference HR (HR*, black dashed line), the measured HR (HR, red line) and the simulated HR response (HR$_{sim}$, black line). The lower plots show the treadmill speed command (black line). The green horizontal line marks the nominal evaluation period for outcome calculation. Pxy denotes the participant number. A: Lowest RMSE for $C_1$, P07. B: Lowest RMSE for $C_2$, P12. C: Middle RMSE for $C_1$, P22. D: Middle RMSE for $C_2$, P14. E: Highest RMSE for $C_1$, P10. F: Highest RMSE for $C_2$, P06.

**Table 1. Outcomes for $C_1$ and $C_2$ and $p$-values for comparison of means (see also Fig 6).**

| | mean ± SD | | MD (95% CI) | $p$-value |
|---|---|---|---|---|
| | $C_1$ | $C_2$ | $C_2 - C_1$ | |
| RMSE/bpm | 2.13 ± 0.35 | 1.98 ± 0.49 | -0.14 (-∞, -0.024) | 0.026 |
| $P_{\nabla u}/(10^{-4}\ \mathrm{m^2/s^2})$ | 20.01 ± 9.85 | 23.37 ± 13.09 | 3.36 (1.05, ∞) | 0.011 |

$n = 20$

SD: standard deviation

MD: mean difference ($C_2 - C_1$)

CI: confidence interval

RMSE: root-mean-square error

$P_{\nabla u}$: average control signal power

bpm: beats per minute

$20.01 \times 10^{-4}\ \mathrm{m^2/s^2} \pm 9.85 \times 10^{-4}\ \mathrm{m^2/s^2}$, $C_2$ vs. $C_1$, $p = 0.011$ (Table 1, Fig 6B). To better demonstrate the performance improvement of $C_2$, a pair of measurements from participant P10 are provided (Fig 7).

## Discussion

The aim of this work was to theoretically derive a generic feedback design strategy that achieves a constant input sensitivity function for LTI plant models, and to empirically test whether a compensator based on a second-order model is more dynamic (higher $P_{\nabla u}$) and has better HR tracking accuracy (lower RMSE) than a compensator based on a first-order model.

The theory for constant input sensitivity feedback design was first derived for general LTI plant models and was then specialised to the first and second-order cases. For the specific compensators employed here, it was found that the input sensitivity functions $U_o$ were indeed constant for all frequencies, and that the sensitivity and complementary sensitivity functions $S_o$

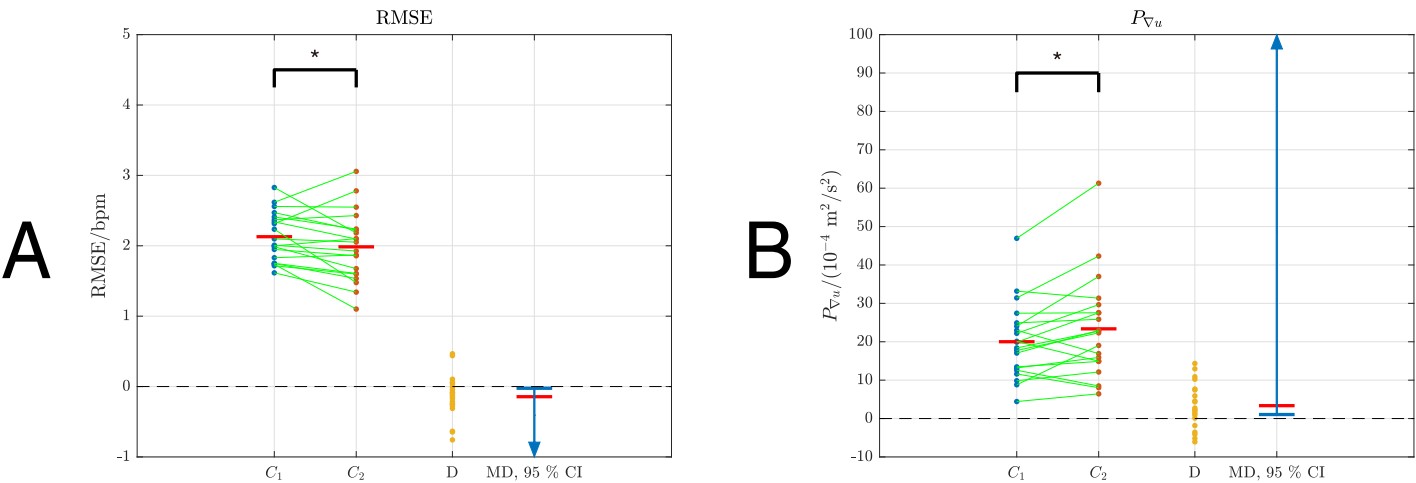

**Fig 6. Graphical depiction of statistical comparison between outcomes.** Blue ($C_1$) and red ($C_2$) dots are the individual samples; green lines connect samples from individual participants; red bars mark the sample means (given numerically in Table 1). D shows the differences between paired samples ($C_2 - C_1$). MD shows mean differences (red horizontal bars) and the corresponding 95% confidence intervals (CIs, blue lines and arrows). When the difference is significant, the value 0 lies outwith the CI; the * notation indicates $p < 0.05$. A: RMSE for $C_1$ and $C_2$, $p = 0.026$. B: $P_{\nabla u}$ for $C_1$ and $C_2$, $p = 0.011$.

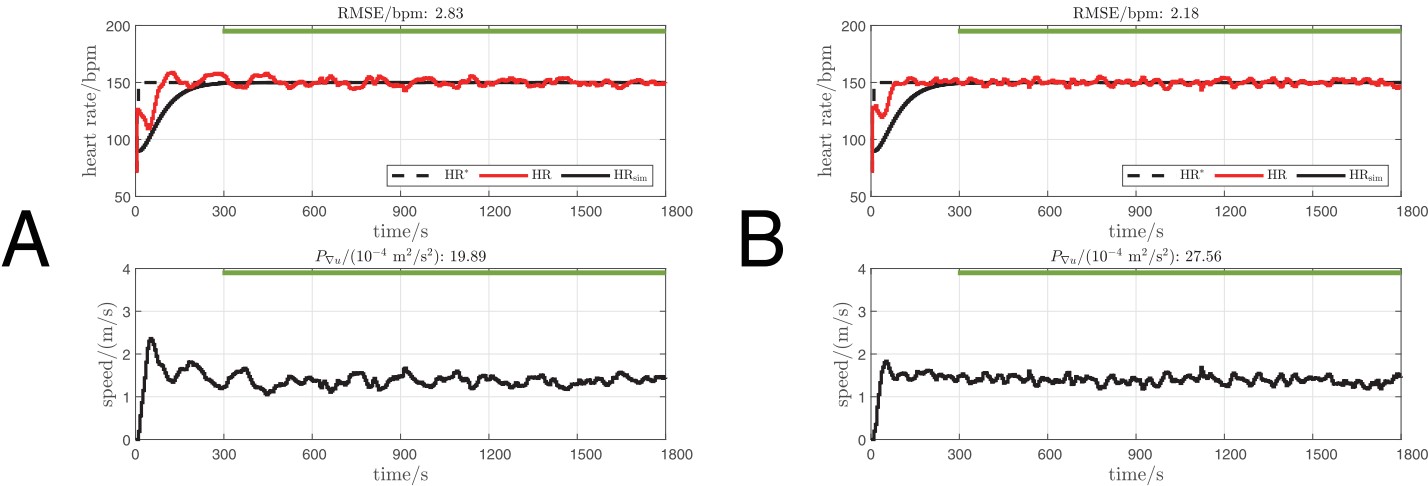

**Fig 7. Measurements with $C_1$ and $C_2$ from participant P10.** Linestyles and legend definitions are the same as in Fig 5. A: Measurement with compensator $C_1$. B: Measurement with compensator $C_2$.

and $T_o$ displayed little or no peaking (Fig 2). Furthermore, the respective closed-loop frequency responses for the $C_1$, $P_1$ and $C_2$, $P_2$ combinations were nominally very similar.

It is notable that the compensator transfer functions for both $C_1$ and $C_2$, Eqs (18) and (24), have a very simple functional form and their coefficients depend only on the respective nominal model parameters (gains and time constants).

Overall, it was found that compensator $C_2$ was significantly more accurate, i.e. it had 7% lower mean RMSE, and significantly more dynamic, i.e. it had 17% higher mean $P_{\nabla u}$, than $C_1$. This improvement likely stems from the substantially and significantly better fidelity of second-order models compared to first order models [17], in line with classical descriptions of the different phases of the cardiac response to exercise [18]. These outcomes, achieved using a treadmill, are consistent with previous observations for the cycle ergometer exercise modality [19].

In summary, whenever heart rate tracking accuracy is of primary importance and a more dynamic control signal is acceptable, the use of a compensator based on a second-order nominal model is recommended.

## Author Contributions

**Conceptualization:** Hanjie Wang, Kenneth J. Hunt.

**Data curation:** Hanjie Wang.

**Formal analysis:** Hanjie Wang, Kenneth J. Hunt.

**Writing – original draft:** Hanjie Wang.

**Writing – review & editing:** Kenneth J. Hunt.

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
