## [Decision Letter · Decision Letter 0]

11 Aug 2023

PONE-D-23-04752Feedback control of heart rate during treadmill exercise based on a two-phase response modelPLOS ONE

Dear Dr. Wang,

Thank you for submitting your manuscript to PLOS ONE. After careful consideration, we feel that it has merit but does not fully meet PLOS ONE’s publication criteria as it currently stands. Therefore, we invite you to submit a revised version of the manuscript that addresses the points raised during the review process.

ACADEMIC EDITOR:Dear Authors,one expert in the field revised your manuscript reporting several minor issues you should consider during the revision process. Please submit your revised manuscript by Sep 25 2023 11:59PM. If you will need more time than this to complete your revisions, please reply to this message or contact the journal office at plosone@plos.org. Please include the following items when submitting your revised manuscript:A rebuttal letter that responds to each point raised by the academic editor and reviewer(s). You should upload this letter as a separate file labeled 'Response to Reviewers'.A marked-up copy of your manuscript that highlights changes made to the original version. You should upload this as a separate file labeled 'Revised Manuscript with Track Changes'.An unmarked version of your revised paper without tracked changes. You should upload this as a separate file labeled 'Manuscript'.If applicable, we recommend that you deposit your laboratory protocols in protocols.io to enhance the reproducibility of your results. Protocols.io assigns your protocol its own identifier (DOI) so that it can be cited independently in the future. For instructions see: https://journals.plos.org/plosone/s/submission-guidelines#loc-laboratory-protocols. Additionally, PLOS ONE offers an option for publishing peer-reviewed Lab Protocol articles, which describe protocols hosted on protocols.io. Read more information on sharing protocols at https://plos.org/protocols?utm_medium=editorial-email&utm_source=authorletters&utm_campaign=protocols.

We look forward to receiving your revised manuscript.

Kind regards,

Emiliano Cè

Academic Editor

PLOS ONE

Journal Requirements:

6. We note that Figure 4 includes an image of a participant in the study. 

Reviewers' comments:

Reviewer's Responses to Questions

**Comments to the Author**

1. Is the manuscript technically sound, and do the data support the conclusions?

Reviewer #1: Yes

2. Has the statistical analysis been performed appropriately and rigorously? 

Reviewer #1: I Don't Know

3. Have the authors made all data underlying the findings in their manuscript fully available?

Reviewer #1: Yes

4. Is the manuscript presented in an intelligible fashion and written in standard English?

Reviewer #1: Yes

5. Review Comments to the Author

Reviewer #1: The manuscript deals with the problem of controlling the heart rate during treadmill exercises via a feedback control law, assuming a linear-time-invariant plant model, such that the tracking of a given reference signal for the heart rate is achieved. The authors investigate the difference between a first-order and a second-order controller to obtain certain requirements for different sensitivity functions. The result is that a controller based on a second-order nominal model is recommended, achieving a better reference tracking accuracy. Finally, data from experiments is provided to illustrate the effectiveness of the proposed solution.

In my opinion, the paper is well written and the results sound solid. Below there is a list of minor things that, in my opinion, should be addressed by the authors to better highlight their results:

1) In the abstract, there are some artifacts, due to an incorrect syntax in referencing to some equations. This should be fixed.

2) The authors say that the controller based on a second-order model (C2) is more "dynamic" with respect to the one based on a first-order model (C1). It is not clear what the authors mean with the term "dynamic". Furthermore, only in the "Experimental results" it is stated that it is related to the control u having an higher "average power of change". This is not clear.

3) In the "Results" section, there are missing references (e.g. Tab. 1, Fig. ???). These error should be fixed.

4) Figure 5 could use better colors/styles for the signals. The lines overlap too much and it is not clear why the authors have used the red color for both the "nominal evaluation period for outcome calculation" and to represent the measured HR. Furthermore, the legend is wrong, because in the text there is no reference to the "HR_{nom}" signal, while it is mentioned a "HR_{sim}".

5) Although I suppose it would have been difficult to achieve, I think that a direct comparison between the performances obtained with the controllers C1 and C2 (e.g. showing both the use of controller C1 and C2 for the same participant), would have been more useful, to better highlight the performance improvement that a second-order controller has over a first-order one.

6. PLOS authors have the option to publish the peer review history of their article (what does this mean?). If published, this will include your full peer review and any attached files.

Reviewer #1: No

---

## [Author Response · Author response to Decision Letter 0]

30 Aug 2023

For details, please check attached "Response to Reviewers.docx".

Editor

RESPONSE: Thank you for your constructive and helpful comments. Please find our point-by-point responses below..

RESPONSE: The template has been applied.

RESPONSE: The ethics statement was moved to the Methods section.

6. We note that Figure 4 includes an image of a participant in the study. 

RESPONSE: The person in Fig. 4 is not a study participant, rather the part of this figure showing the treadmill and runner was provided to us with explicit approval from the treadmill manufacturer h/p/cosmos – this forms part of their product literature. To prevent identification of this person, their face has now been blurred.

RESPONSE: The reference list has been reviewed as instructed.

Reviewer #1

RESPONSE: Thank you for your constructive and helpful comments. Please find below a point-by-point response. Changes to the manuscript have been highlighted in red font.

1) In the abstract, there are some artifacts, due to an incorrect syntax in referencing to some equations. This should be fixed.

RESPONSE:

This has been corrected.

2) The authors say that the controller based on a second-order model (C2) is more "dynamic" with respect to the one based on a first-order model (C1). It is not clear what the authors mean with the term "dynamic". Furthermore, only in the "Experimental results" it is stated that it is related to the control u having an higher "average power of change". This is not clear.

RESPONSE:

To clarify this point, the following text has been added to the Introduction (lines 79-82):

“The concept of a controller being more or less dynamic is quantified in this work using the average power of changes in the control signal (the treadmill speed command; see Eq. (32) in the sequel).”

This outcome, the average power of changes in the control signal, was already explicitly defined in the Methods, viz. in Eq. (32).

3) In the "Results" section, there are missing references (e.g. Tab. 1, Fig. ???). These error should be fixed.

RESPONSE:

This has been corrected.

4) Figure 5 could use better colors/styles for the signals. The lines overlap too much and it is not clear why the authors have used the red color for both the "nominal evaluation period for outcome calculation" and to represent the measured HR. Furthermore, the legend is wrong, because in the text there is no reference to the "HR_{nom}" signal, while it is mentioned a "HR_{sim}".

RESPONSE:

1. We have changed the colour of the horizontal bar indicating the evaluation period to GREEN, to avoid any conflict with the red colour used for HR.

2. Given that the controller aims to ensure that the measured HR follows the target and nominal/simulated HR, and since the controllers in this work are very accurate, it is inevitable that there will be signal overlap. To make things clearer, however, we have replotted these graphs to bring the red lines for measured HR to the forefront of the plots.

3. The legends have all been corrected to use only “sim” as in the corresponding Eq. (31).

5) Although I suppose it would have been difficult to achieve, I think that a direct comparison between the performances obtained with the controllers C1 and C2 (e.g. showing both the use of controller C1 and C2 for the same participant), would have been more useful, to better highlight the performance improvement that a second-order controller has over a first-order one.

RESPONSE:

We have added a new figure – Fig. 7 – that shows a direct comparison of C1 and C2 for the same participant – participant P10. As suggested, this highlights the performance improvement for C2 vs. C1. The following text has been added to the Results (lines 285-286):

“To better demonstrate the performance improvement of C2, a pair of measurements from participant P10 are provided (Fig 7).”

---

## [Decision Letter · Decision Letter 1]

19 Sep 2023

Feedback control of heart rate during treadmill exercise based on a two-phase response model

PONE-D-23-04752R1

Dear Dr. Wang,

We’re pleased to inform you that your manuscript has been judged scientifically suitable for publication and will be formally accepted for publication once it meets all outstanding technical requirements.

Kind regards,

Emiliano Cè

Academic Editor

PLOS ONE

Additional Editor Comments (optional):

Reviewers' comments:

Reviewer's Responses to Questions

**Comments to the Author**

1. If the authors have adequately addressed your comments raised in a previous round of review and you feel that this manuscript is now acceptable for publication, you may indicate that here to bypass the “Comments to the Author” section, enter your conflict of interest statement in the “Confidential to Editor” section, and submit your "Accept" recommendation.

Reviewer #1: All comments have been addressed

2. Is the manuscript technically sound, and do the data support the conclusions?

Reviewer #1: (No Response)

3. Has the statistical analysis been performed appropriately and rigorously? 

Reviewer #1: (No Response)

4. Have the authors made all data underlying the findings in their manuscript fully available?

Reviewer #1: (No Response)

5. Is the manuscript presented in an intelligible fashion and written in standard English?

Reviewer #1: (No Response)

6. Review Comments to the Author

Reviewer #1: (No Response)

7. PLOS authors have the option to publish the peer review history of their article (what does this mean?). If published, this will include your full peer review and any attached files.

Reviewer #1: No

---

## [Editor Report · Acceptance letter]

12 Oct 2023

PONE-D-23-04752R1 

Feedback control of heart rate during treadmill exercise based on a two-phase response model 

Dear Dr. Wang:

I'm pleased to inform you that your manuscript has been deemed suitable for publication in PLOS ONE. Congratulations! Your manuscript is now with our production department. 

Kind regards, 

on behalf of

Prof. Emiliano Cè 

Academic Editor

PLOS ONE